# SCREEN: A Benchmark for Situated Conversational Recommendation

Dongding Lin*
The Hong Kong Polytechnic University
Hong Kong, China
dongding88.lin@connect.polyu.hk

Jian Wang*
The Hong Kong Polytechnic University
Hong Kong, China
jian-dylan.wang@connect.polyu.hk

Chak Tou Leong
The Hong Kong Polytechnic University
Hong Kong, China
chak-tou.leong@connect.polyu.hk

Wenjie Li
The Hong Kong Polytechnic University
Hong Kong, China
cswjli@comp.polyu.edu.hk

## Abstract

Engaging in conversational recommendations within a specific scenario represents a promising paradigm in the real world. Scenario-relevant situations often affect conversations and recommendations from two closely related aspects: varying the appealingness of items to users, namely *situated item representation*, and shifting user interests in the targeted items, namely *situated user preference*. We highlight that considering those situational factors is crucial, as this aligns with the realistic conversational recommendation process in the physical world. However, it is challenging yet under-explored. In this work, we are pioneering to bridge this gap and introduce a novel setting: *Situated Conversational Recommendation Systems* (SCRS). We observe an emergent need for high-quality datasets, and building one from scratch requires tremendous human effort. To this end, we construct a new benchmark, named **SCREEN**, via a role-playing method based on multimodal large language models. We take two multimodal large language models to play the roles of a user and a recommender, simulating their interactions in a co-observed scene. Our SCREEN comprises over 20k dialogues across 1.5k diverse situations, providing a rich foundation for exploring situational influences on conversational recommendations. Based on the SCREEN, we propose three worth-exploring subtasks and evaluate several representative baseline models. Our evaluations suggest that the benchmark is high quality, establishing a solid experimental basis for future research. The code and data are available at https://github.com/DongdingLin/SCREEN.

## CCS Concepts

• **Computing methodologies** → **Discourse, dialogue and pragmatics**; • **Information systems** → **Recommender systems**.

## Keywords

Benchmark; Situated Conversational Recommendation; Role-playing

---

*Both authors contributed equally to this research.

**ACM Reference Format:**
Dongding Lin, Jian Wang, Chak Tou Leong, and Wenjie Li. 2024. SCREEN: A Benchmark for Situated Conversational Recommendation. In *Proceedings of the 32nd ACM International Conference on Multimedia (MM '24), October 28-November 1, 2024, Melbourne, VIC, Australia.* ACM, New York, NY, USA, 10 pages. https://doi.org/10.1145/3664647.3681651

## 1 Introduction

Building a Conversational Recommendation System (CRS) [4, 14, 22] that can communicate with people in multimodal situations is an attractive goal for the AI community. Existing multimodal CRSs [7, 27, 38, 40, 44] integrate textual and visual product information in various ways to enhance recommendation processes. Chen et al. [3], Nie et al. [29] enrich the multimodal context by incorporating item images and dialogue histories. Meanwhile, Zhang et al. [42] employs a multi-attribute graph model to capture diverse item attributes. Additionally, Du et al. [6] advances item representations through a multimodal transformer, capturing both global and local perspectives of items. Multimodal CRSs are targeted to integrate textual and visual information to model user preferences and item representations, expecting the system to provide precise and appropriate recommendations.

Despite considerable advancements, existing multimodal CRSs still face challenges in understanding dynamic user preferences and accurately representing real-world items. This is mainly due to two key aspects: (1) User Preference Modeling: current methods [7, 38, 44] narrowly leverage a user's historical profiles, general interests, and conversational histories to model user preferences. These approaches typically overlook the dynamic nature of user interests and choices, which can fluctuate significantly due to situational factors, such as product location and the current season's climate. This leads to what can be termed as *situated user preference* — a concept that requires grounding the user's underlying interests to the current environment and situational context. (2) Item Representation: conventional modeling in CRSs often represents items through intrinsic and static attributes [6]. It fails to account for the variability in an item's attraction, which can change with situational factors like the spatial layout and daily weather, leading to what can be termed as *situated item representation*. This concept extends beyond traditional static attributes by adapting to environmental contexts, providing a more context-sensitive item representation. As demonstrated in Figure 1, considering the situation (e.g., spring) beyond the intrinsic attributes of an item (e.g.,

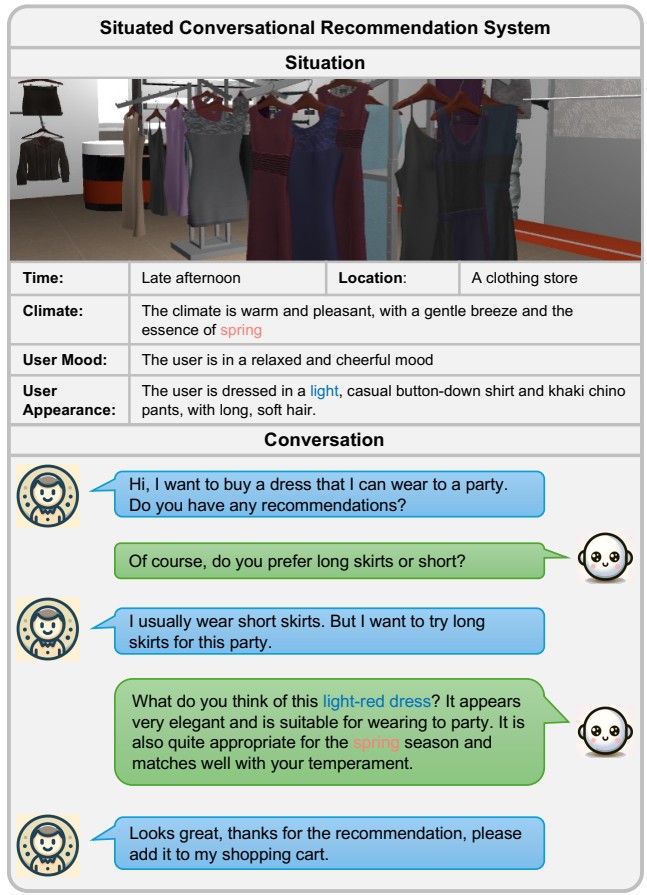

**Figure 1: An illustrative example for Situated Conversational Recommendation System (SCRS).**

brand or appearance) leads to a more accurate representation of the item in the given context. Moreover, by analyzing the user's outfit (e.g., dressed in a light shirt), emotion (e.g., cheerfulness), and dialogue history, the system captures user preferences more accurately within the current scenario. Afterward, the system is able to recommend a personalized product (e.g., a light-red dress) that is more appealing to the user.

In this work, we extend the traditional multimodal CRS to a more realistic paradigm: **S**ituated **C**onversational **R**ecommendation **S**ystem (**SCRS**). It requires the system to consider the inherent connection between users and items under specific situational contexts, thereby conversing with higher engagement and providing more appropriate recommendations to users. While the advancement of SCRS is essential, the absence of a high-quality benchmark remains a significant obstacle to its development. Thus, we raise the question: *how can we utilize minimal human efforts to construct a high-quality SCRS dataset?*

Inspired by the powerful human-mimicking capabilities of large language models (LLMs) [10, 19], we construct a high-quality SCRS benchmark, namely **S**ituated **C**onversational **RE**comm**EN**dation (**SCREEN**), based on a role-playing approach. We use carefully designed instructions to prompt LLMs as agents to play the roles

of users and recommenders in a simulated environment. To let them "see" the co-observed scene, we use multimodal LLMs to extract visual features given specific situations. In the end, we obtain over 1.5k scenes with 20k recommendation-oriented dialogues. Moreover, we delineate three critical subtasks to evaluate SCRS comprehensively: *system action prediction*, *situated recommendation*, and *system response generation*. These subtasks measure the system's performance in accurately interpreting user intentions, modeling situated item representations, capturing situated user preferences, and generating responses that actively engage users. Additionally, we employ multiple representative baseline models and evaluate their performances on the SCREEN benchmark.

Our contributions are summarized as follows: (1) We expand the scope of traditional multimodal CRS to SCRS. This under-explored yet promising paradigm incorporates situational context into the recommendation reasoning process, furnishing users with more engaging and contextually appropriate recommendations. (2) We construct a comprehensive and high-quality benchmark named SCREEN to facilitate exploration in this nascent field. (3) We identify and articulate three essential subtasks for evaluating SCRS. We further present baseline results on the SCREEN, establishing a solid experimental basis for future research.

## 2 Related Work

### 2.1 Conversational Recommendation Systems

Conversational recommendation systems (CRS) have become a major research focus, delivering superior recommendations through natural language interactions [14, 22, 34]. Most CRS datasets, including REDIAL [20], TG-REDIAL [46], INSPIRED [11], and DuRecDial [23, 24], rely heavily on text, using dialogue histories and item attributes but neglect the crucial role of visual information associated with items. To address the need for multimodal CRS, the introduction of the MMD benchmark dataset [33] marked a significant advancement, initiating tasks that cater to multimodal, domain-specific dialogues. The MMConv dataset [21] further expanded this by covering multiple domains. Despite these advancements, existing datasets fail to fully capture the diverse expressions of users' subjective preferences and recommendation behaviors in real-life scenarios, a gap the SURE dataset [25] seeks to fill. The SIMMC-VR dataset [36] also enhances the system's comprehension of spatial and temporal contexts. However, integrating situational context into CRS—adapting recommendations based on users' environments and activities—remains underexplored, presenting a promising direction for developing more context-aware systems.

### 2.2 Situated Dialogues

Recent advancements in situated dialogues have emphasized the importance of embedding interactions within specific contextual situations, driving interest in training agents for multimodal actions grounded in dynamic multimodal input and historical dialogue context. To facilitate this research, Crook et al. [5], Moon et al. [28] developed the SIMMC dataset, establishing a foundation for situational, interactive multimodal conversations. Despite its significance, the SIMMC dataset faced criticism for its simplistic and unrealistic multimodal contexts. To this end, Kottur et al. [17] introduced SIMMC 2.0, enhancing multimodal dialogue capabilities

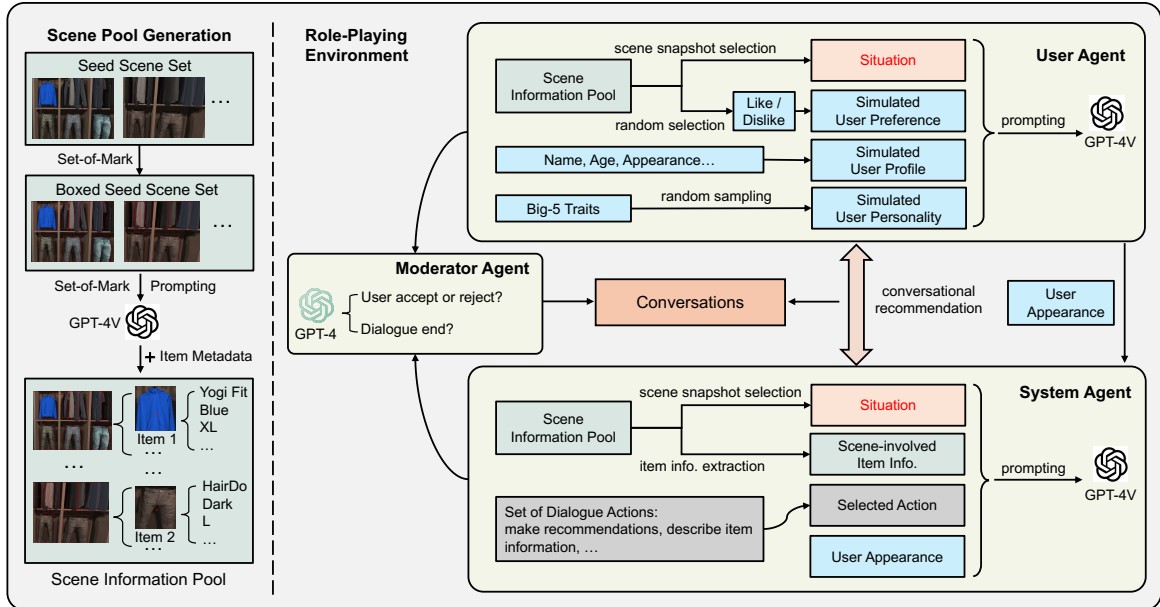

**Figure 2: Overview of our automatic dataset construction framework for situated conversational recommendation.**

but focusing primarily on immediate, local topics, which limited its support for more dynamic, forward-looking conversations. To address this limitation, Otani et al. [31] presented SUGAR, a dataset to improve agents' proactive response selection. Despite these advancements, integrating situated conversations into CRSs presents untapped potential. It demands a deep understanding of user preferences and item representations in specific contexts. Our work incorporates situational factors into recommendation reasoning, enabling the system to deliver more accurate, contextually relevant recommendations that align with the user's situational context and preferences, potentially enhancing user engagement.

## 3 SCRS Dataset Construction

### 3.1 Problem Formulation

We consider a SCRS dataset $\mathcal{D} = \{(\mathcal{S}_i, \mathcal{I}_i, \mathcal{U}_i, C_i)\}_{i=1}^N$, where $N$ is the total number of dialogues. In the $i$-th dialogue, $\mathcal{S}_i$ represents the situational information, which includes the user-system co-observed scenario (e.g., scene snapshot), spatiotemporal information (e.g., time), and environmental information (e.g., climate). $\mathcal{I}_i$ represents all items in this situation $\mathcal{S}_i$. $\mathcal{U}_i$ denotes the user's personalized information (e.g., user profile, user state). $C_i = \{C_{i,t}\}_{t=1}^{N_T}$ is the dialogue content, with a total of $N_T$ turns. The task of SCRS is formalized as follows: given the situational information $\mathcal{S}$, all items $\mathcal{I}$ in this situation, a set of user's personalized information $\mathcal{U}$, and a dialogue context $C$, the objective is to select and recommend the most appropriate item in the scene $\mathcal{S}$ to the user and generate a natural language response that matches the scene content. Compared with traditional CRS, SCRS requires that the recommended items and responses closely relate to the situational context.

This section describes a role-playing framework for constructing an SCRS dataset by integrating multiple LLM agents, inspired by

**Scene Information Pool Instruction Template:**
Imagine yourself as a consumer viewing a scene screenshot. Several boxes are drawn on the image, and each box encloses a piece of clothing or furniture, marked with a corresponding number. Describe them according to the number sign, focusing on the color, type, and pattern. Below is the name of each piece of clothing or furniture. It is auxiliary information to help you identify them:
1. <NAME_OF_ITEM 1>
2. <NAME_OF_ITEM 2>
…

**Figure 3: Instruction template for the scene pool generation.**

[35]. As depicted in Figure 2, the framework comprises two key components: generating a *Scene Information Pool* and establishing a *Role-Playing Environment*. First, scene item metadata only describes its intrinsic attributes, lacking the subjective descriptions common in real-world user-salesperson interactions [25]. To address this, we use multimodal LLMs to generate subjective descriptions, enriching item information (e.g., describing a clothing item as "enthusiastic and bold" rather than just "very red"). Second, the role-playing environment includes three agents: *user*, *system*, and *moderator*, each following meticulously designed instructions to facilitate effective communication and interaction. Our work leverages VR snapshots from the SIMMC 2.1 dataset [16], encompassing diverse scenes from 140 fashion and 20 furniture stores, with detailed metadata covering nine attributes per item, including *type, color, pattern, material, price, brand, size, sleeve length*, and *consumer reviews*.

### 3.2 Scene Information Pool Generation

When making conversational recommendations in scenarios, users often prioritize a product's situational attributes (e.g., appearance

and location) over intrinsic ones (e.g., price, brand), leading to more intuitive decision-making. Users tend to prejudge products based on these situational attributes, which can vary depending on the scenario and be influenced by external factors like lighting or item placement. For example, pants may appear more vibrant under soft lighting than under bright lighting. While existing research typically describes products using specific referring expressions (e.g., "red clothes"), non-expert users often use subjective descriptions (e.g., "clothes designed for young women"), which are generally missing from conventional product meta-databases. To address this, we enhance item metadata with situational attributes and subjective descriptors, providing a more nuanced item representation. Leveraging recent advancements in multimodal LLMs, particularly the Set-of-Mark technique [39], we improve item recognition and description generation. Using spatial data from the SIMMC 2.1 dataset, which provides precise coordinates for products within scene snapshots, we create bounding boxes and assign unique identifiers for each item. These annotated snapshots are processed by GPT-4V (`gpt-4-1106-vision-preview` version), tasked with elucidating situational attributes and subjective descriptions based on the prompt shown in Figure 3. The output is integrated into existing product metadata to form a comprehensive scene information pool.

## 3.3 Role-Playing Environment

Our role-playing environment is crafted to provide a global environment description that prompts all LLM agents. To engender a realistic and multifaceted setting, it incorporates three principal dimensions: (1) *Temporal phases*, which are delineated into morning, noon, afternoon, and evening; (2) *Spatial settings*, which encompass both fashion and furniture retail spaces; and (3) *Climate*, which is represented by the quartet of seasons: spring, summer, autumn, and winter. To augment the diversity within the simulation, we employ ChatGPT (`gpt-3.5-turbo` version) to generate succinct narratives for each seasonal context. For example, in the scenario "*afternoon, fashion store, spring*," the ambiance is vividly depicted as "*It is the afternoon, and you find yourself in a fashion store. A gentle breeze wafts through, heralding the arrival of spring.*" Such tailored descriptions are appended to the beginning of each agent's instructions, ensuring a coherent framework for interaction.

*3.3.1 User Agent.* The user agent primarily aims to simulate consumers' shopping behavior across diverse scenarios, generating responses based on their preferences, profiles, and personalities. To this end, we set user information through the following aspects:

*User Preference.* In the given scenario, we catalog the attributes of all products and allocate user preferences (favor, aversion, or neutrality) to each attribute randomly. This approach facilitates the generation of a wide range of personalized preferences. To enrich the expression of user preferences and inject it with a more natural and diverse vocabulary, we employ ChatGPT to refine this structured information into fluent natural language. Figure 4 illustrates the instruction template used for this transformation. Subsequently, structured user preferences are described more naturally, such as, "*You exhibit a preference for red, an aversion to white, and display no particular inclination towards purple;...*"

---

**User Preference Instruction Template:**
The following is a structured expression of user preference. Please refine this structured information in natural and fluent language. Be careful to start with "you," and the length should not exceed 50 words.
Color: red (favor), white (aversion), purple (neutral)…
Style: jacket (favor), shirt (favor), sweater (aversion)…
…

**User Profile Instruction Template:**
The following is a structured expression of a user profile. Please refine this structured information into natural and fluent language. Be careful to start with "you," and the length should not exceed 50 words.
Name: John; Age: 18 years old; Gender: Male; Profession: Doctor; Emotional State: Joyful; Upper Body: White shirt; Lower Body: Jeans; Hair Style: Short; …

**User Personality Instruction Template:**
The following is a structured expression of a user's personality. Please refine this structured information into natural and fluent language. Be careful to start with "you," and the length should not exceed 50 words.
Openness: Intellectual; Conscientiousness: Efficient; …

**Figure 4: Instruction template for the user preference, user profile, and user personality generation.**

---

*User Profile.* Leveraging the user information in the DuRecDial dataset [23], we developed a structured pool of personal profile attributes, including but not limited to name, age, gender, and profession. For instance, a typical user profile in this structured pool might be described as follows: "*Name: John; Age: 18; Gender: Male; ...*". Additionally, we enriched these profiles with emotional states (e.g., joy, cheerfulness, excitement, sadness, worry, and grief) and appearance descriptions based on items captured in another scene snapshot to mirror real-user scenarios. An example could be "*Emotional State: Joyful; Upper Body: White shirt; Lower Body: Jeans; ...*". It is crucial to highlight that, similar to how a salesperson makes recommendations based on the user's appearance in real life, the system agent can also observe the user's appearance to infer the user's preferences and make appropriate recommendations. We use ChatGPT to refine this structured information into fluent natural language similar to processing user preferences information. The instruction template is shown in Figure 4.

*User Personality.* To further reflect the user's personality and increase the user agent's diversity, we also use the Big Five personality traits [9, 41] to simulate user personalities. These traits provide a framework for the assignment of attributes representing positive and negative aspects along five dimensions: openness (O), conscientiousness (C), extraversion (E), agreeableness (A), and neuroticism (N). Combining these characteristics allows for creating a nuanced and comprehensive user personality model, enriching diverse interactions. As shown in Figure 4, we leverage ChatGPT to refine such structured information into fluent natural language, similar to how user preferences are processed.

In the end, we use natural languages to express the simulated user and prompt the user agent to play the role of a customer. Figure 5 shows the complete instruction template.

**User Agent Instruction Template:**
Imagine yourself as a consumer shopping at a clothing store | furniture store. This image is a snapshot of the store. Here are your details:
1. <Generated User Preference>
2. <Generated User Profile>
3. <Generated User Personality>
You need to judge whether the system's recommendations align with your criteria. Your response should be concise, no more than 50 words. You do not need to recommend anything but feel free to express your interests.

- - - - - - - - - - - - - - - - - - - - - - - -

**System Agent Instruction Template:**
Imagine yourself as a salesperson in a clothing store | furniture store. This image is a snapshot of the store. The following is the customer's appearance: <Generated User Appearance>.You have metadata and subjective description for all items in the store: <Item Metadata>; <Generated Item Subjective Description>. You need to first choose one of the 6 actions (<Predefined Actions>) to decide your next action, and generate a corresponding response based on your action. Please output in the format of [Action]:[Reply]. Your response should be concise, no more than 100 words.

- - - - - - - - - - - - - - - - - - - - - - - -

**Moderator Agent Instruction Template:**
You are the moderator of a conversation. You need to decide whether the conversation should end immediately. The conversation should be terminated in the following three situations: (1) The system completes the recommendation, the user accepts the recommendation based on the preset preferences, and the system action is not <Topic Transfer>. (2) The user rejects the system's recommendations multiple times (more than three times) (3) The conversation between the user and the system reaches the maximum number of rounds limit (30 rounds). Should the following dialogue <Ongoing Conversation> be ended? Answer yes or no.

**Figure 5: Instruction template for different agents.**

*3.3.2 System Agent.* The system agent aims to serve as a human-like salesperson, such as a clothing salesperson in a fashion store. Its primary objective is to recommend the most appropriate items based on the user's preferences expressed during the conversation. To realize this vision, we design the system agent with predefined actions: (1) **Describe Item Information**. The system agent proactively offers the user comprehensive details of the items, including intrinsic attributes, situational attributes, and subjective descriptions. (2) **Inquire About Preferences**. The system agent gathers user preferences by querying their opinions on specific items within the scene or clarifying ambiguities in the user's requests to ascertain their needs accurately. (3) **Address User Queries**. The system agent provides the requested information upon user inquiries about an item, ensuring that user inquiries are promptly and effectively addressed. (4) **Topic Transfer**. When the user accepts an item the system agent recommends, the system agent determines whether to introduce another item or to delve deeper into the current selection, thus guiding the conversation strategically. (5) **Make Recommendations**. When the system agent deems sufficient information on user preferences has been collected, it will decide which item to recommend. (6) **Add to Cart**. When the user accepts a recommendation, the system agent inquires whether the user wishes to add the item to their shopping cart. It is worth noting that in each interaction, the system agent is required to identify the action it intends to execute initially. Subsequently, it generates a response that aligns with the specified action.

In addition, similar to real-life shopping experiences, the salesperson can observe the customer's appearance but cannot obtain the customer's profile (e.g., name, profession). In the simulated conversation between the user and the system, the system can get the user's appearance but not the user's private profile. Therefore, we convey information about the user's appearance to the system agent, aiding in understanding and capturing user preferences. In practice, we further enhance system agents with self-augmented instructions, where the agent's prompts will be repeated in each conversation round to avoid forgetting the items' details. The specific system agent instruction template is shown in Figure 5.

*3.3.3 Moderator Agent.* The moderator agent is designed to automatically manage whether the conversation between the system agent and the user agent should be terminated. It also tracks whether the user agent accepts or rejects the recommended items based on their preset preferences. To ensure that the constructed data meets the desired characteristics, we set certain natural language conditions to terminate the conversation. These conditions are summarized as follows: (1) The system agent completes the recommendation, the user agent accepts it, and the recommended item aligns with the agent's predefined preferences. In addition, the system action is not topic transferred. (2) The user agent rejects the recommended items by the system agent multiple times (e.g., more than three times). (3) The interaction is deemed concluded once the conversation between the system agent and the user agent hits the maximum number of turns. Note that the synthesized conversation terminated under the first condition is accepted as valid data, while those that end under the second and third conditions are categorized as invalid and discarded. Figure 5 describes the specific moderator agent instruction template.

### 3.4 Dataset Construction

In this study, the unique multimodal context of our conversational scenario integrates both visual (i.e., scene snapshots) and textual (including dialogue history and instructions) elements. To accommodate this complexity, the user and system agents are powered by GPT-4V (`gpt-4-1106-vision-preview` version), a variant of ChatGPT specially enhanced for multimodal tasks. Conversely, the moderator agent, which functions without reliance on visual cues, utilizes GPT-4 (`gpt-4-1106-preview` version) to navigate its decision-making processes effectively. The dialogue initiation occurs as the system agent greets the user agent, triggering a sequence of interactions that evolve through numerous dialogue rounds. These interactions are concluded ultimately with an intervention from the moderator agent. Collectively, these agents can collaborate to construct large-scale, high-quality dialogues rapidly, significantly reducing the need for human intervention.

Our role-playing framework is built upon the open-source library ChatArena [37]. We have standardized the response generation across all agents by setting a temperature of 0.8. The maximum generation tokens are also tailored for each agent type, with limits set at 120, 80, and 20 for the system, user, and moderator agents,

**Table 1: Comparison between our SCREEN dataset and other related datasets (SB: situation-based, SR: situated recommendation, \*: item images, †: scene snapshots).**

| Dataset | Task | Modality | Participants | SB | SR | Domains | #Image | #Dialogue |
|---------|------|----------|--------------|----|----|---------|--------|-----------|
| REDIAL [20] | CRS | Textual | Crowd Workers | ✗ | ✗ | Movie | - | 10,006 |
| TG-REDIAL [46] | CRS | Textual | Crowd Workers | ✗ | ✗ | Movie | - | 10,000 |
| INSPIRED [11] | CRS | Textual | Crowd Workers | ✗ | ✗ | Movie | - | 1,001 |
| MMD [33] | Multimodal CRS | Textual+Visual | Crowd Workers | ✗ | ✗ | Fashion | 4,200* | 105,439 |
| SIMMC 2.0 [17] | Situated Dialogue | Textual+Visual | Crowd Workers | ✓ | ✗ | Fashion, Furniture | $1,566^\dagger$ | 11,244 |
| SURE [25] | Multimodal CRS | Textual+Visual | Crowd Workers | ✓ | ✗ | Fashion, Furniture | $1,566^\dagger$ | 12,180 |
| SCREEN | Situated CRS | Textual+Visual | LLM agents | ✓ | ✓ | Fashion, Furniture | $1,566^\dagger$ | 20,112 |

**Table 2: Statistics of the SCREEN dataset.**

| | |
|---|---|
| Total #dialogue(train/valid/test) | 16,089/2,011/2,012 |
| Total #utterances(train/valid/test) | 172,152/20,713/21,528 |
| Total #scene snapshots | 1,566 |
| Avg. #words per user turns | 15.7 |
| Avg. #words per assistant turns | 20 |
| Avg. #utterances per dialog | 10.7 |
| Avg. #objects mentioned per dialog | 4.3 |
| Avg. #objects in scene per dialog | 19.7 |

respectively. This structured approach ensures a balanced and efficient dialogue generation process, catering to the distinct needs of each agent's role in the conversational architecture.

## 4 SCREEN Dataset

Based on our dataset construction framework, we build a high-quality SCRS dataset named **SCREEN**. Compared to related multimodal CRS datasets, our SCREEN uniquely targets situated recommendations. We first provide a comprehensive overview of the SCREEN dataset, then propose three essential sub-tasks to measure SCRS, including the task formulation and evaluation metrics.

### 4.1 Dataset Statistics

Table 1 presents a comparative analysis between the SCREEN dataset and other related datasets. To our knowledge, the SCREEN dataset is the first dataset within the SCRS domain designed to facilitate recommendations in distinct scenarios. In deviation from conventional text-based CRS datasets, such as REDIAL, SCREEN incorporates visual elements, thus enabling a more comprehensive modeling of item representations. While datasets like MMD and SURE also integrate visual information, they do not consider user preferences and item representations in specific scenarios. SIMMC 2.0 serves as a task-oriented dataset geared towards situational dialogues, while the SCREEN dataset distinguishes itself by concentrating on situated conversational recommendations. Moreover, the SCREEN dataset's inclusion of detailed, personalized information—namely, the Big-5 personality traits—in creating user agents ensures that the generated utterances are more natural and realistic.

Table 2 presents a comprehensive analysis of the SCREEN dataset. As the table delineates, the dataset is divided into training, validation, and test sets, adhering to an 8:1:1 ratio. A notable observation

is that sentences generated by the system agent are longer than those generated by the user agent. This discrepancy stems from the system agent's necessity to introduce detailed information regarding the items to the user. On average, each dialogue mentions approximately four distinct objects, while each conversational scene involves around 20 objects. A particular feature of the SCRS, as opposed to traditional CRS, is that each dialogue within SCRS is associated with a unique list of recommendation candidates, diverging from the conventional approach where all conversations access a communal candidate list. Consequently, the SCRS framework is required to model the representation of items within the conversational scene to provide appropriate recommendations.

### 4.2 Task Formulation

We delineate three sub-tasks to explore the performance of SCRS: *system action prediction*, *situated recommendation*, and *system response generation*. These subtasks are essential to validate whether an SCRS comprehends the situation, user intent, and conversational history nuancedly, which are critical for delivering accurate and contextually relevant recommendations. The system action prediction measures the ability to generate guided actions to satisfy user needs. The situated recommendation measures how well the situational context is utilized to tailor recommendations to user preferences. The system response generation focuses on crafting natural, coherent, and context-aware responses crucial for sustaining user engagement and satisfaction.

*4.2.1 System Action Prediction.* As delineated in Section 3.3.2, the system agent determines its subsequent actions based on information derived from the dialogue history and the contextual scenario involving the user. This necessitates the system's capability to comprehend the user's intent, capture the user's preferences, and incorporate the attributes of items present within the scenario to decide the next step (e.g., make recommendations). The system's performance is quantitatively assessed by calculating the aggregate precision, recall, and F1 scores of the system's action predictions.

*4.2.2 Situated Recommendation.* Building upon the foundation laid by [17], we extend the traditional tasks in CRS to encompass the situated recommendation task as a principal task within SCRS. This pivotal task requires the system to align items' attributes with the user's situated preferences, leveraging the scenario, dialogue history, and detailed item information to deduce the aptest recommendation for the user. It is important to note that a recommendation

**Table 3: Automatic evaluation results of representative baseline models on three proposed subtasks based on the SCREEN dataset. The best results are highlighted in bold ($t$-test with $p$-value $< 0.05$).**

| Model | System Action Prediction | | | Situated Recommendation | | | System Response Generation | | | | |
|---|---|---|---|---|---|---|---|---|---|---|---|
| | Precision | Recall | F1 | R@1 | R@2 | R@3 | PPL ($\downarrow$) | BLEU-2 | BLEU-3 | DIST-1 | DIST-2 |
| SimpleTOD+MM [5] | 0.715 | 0.736 | 0.725 | 0.085 | 0.161 | 0.244 | 19.3 | 0.089 | 0.041 | 0.028 | 0.114 |
| Multi-Task Learning [18] | 0.727 | 0.753 | 0.740 | 0.107 | 0.199 | 0.298 | 17.5 | 0.105 | 0.054 | 0.031 | 0.112 |
| Encoder-Decoder [12] | 0.838 | 0.856 | 0.847 | 0.148 | 0.277 | 0.425 | 12.7 | 0.140 | 0.071 | 0.038 | 0.178 |
| Reasoner [26] | 0.902 | 0.925 | 0.913 | 0.190 | 0.395 | 0.588 | 10.2 | 0.181 | 0.078 | 0.043 | 0.192 |
| MiniGPT4 [47] | 0.946 | 0.951 | 0.948 | 0.234 | 0.498 | 0.697 | **4.31** | 0.252 | 0.117 | 0.081 | 0.310 |
| GPT-4o [13] | **0.951** | **0.974** | **0.962** | **0.284** | **0.557** | **0.751** | - | **0.276** | **0.132** | **0.107** | **0.337** |

**Table 4: Human evaluation results of baseline models on the SCREEN dataset. "SR" denotes "Situation Relevance", "Inform." denotes "informativeness", $\kappa$ denotes kappa.**

| Model | SR | $\kappa$ | Fluency | $\kappa$ | Inform. | $\kappa$ |
|---|---|---|---|---|---|---|
| SimpleTOD+MM [5] | 0.74 | 0.42 | 1.31 | 0.41 | 0.89 | 0.48 |
| Multi-Task Learning [18] | 0.98 | 0.48 | 1.35 | 0.45 | 1.01 | 0.56 |
| Encoder-Decoder [12] | 1.04 | 0.51 | 1.57 | 0.47 | 1.17 | 0.51 |
| Reasoner [26] | 1.19 | 0.47 | 1.61 | 0.52 | 1.48 | 0.48 |
| MiniGPT4 [47] | 1.42 | 0.55 | 1.91 | 0.52 | 1.70 | 0.49 |
| GPT-4o [13] | **1.50** | 0.50 | **1.95** | 0.49 | **1.75** | 0.52 |

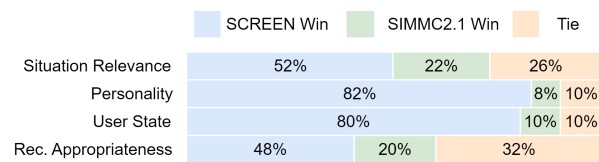

**Figure 6: Human evaluation results of dataset comparison. "Rec." denotes "Recommendation".**

is not mandated at each interaction, as users might seek insights into the item's attributes or other related information. Recommendations are thus made only when the system's action is explicitly to "make recommendations."

Following existing studies [20, 46], we adopt a widely recognized metric for assessing the performance of recommendations. The automatic metric, Recall@$k$ (R@$k$, where $k$= 1, 2, 3), evaluates the accuracy of the top-$k$ items recommended by the model against the ground truth items provided by the system agent.

*4.2.3 System Response Generation.* The objective of this subtask is to generate responses in natural language. The system is required to generate responses based on its decided actions, the historical context of the dialogue, snapshots of the scene, and information of items within the given scenario. Similar to [17], our evaluation metrics include Perplexity (PPL) [15], BLEU-2,3 [32], and Distinct $n$-gram (DIST- $n$, $n$ = 1, 2) [2], to assess the quality of the responses generated by the system. Perplexity serves as an indicator of natural language fluency, wherein a lower Perplexity value signifies a higher degree of fluency. The BLEU-2,3 metric evaluates the concordance of word sequences between the generated responses and the reference responses, with higher BLEU scores indicating closer approximation to the reference responses. The DIST-$n$ measures the diversity of the generated responses at the sentence level, with higher scores denoting a broader variety in sentence construction.

## 5 Experiments

### 5.1 Baseline Models

We implemented and assessed several multimodal baseline models on our proposed SCREEN dataset: (1) **SimpleTOD+MM** Model [5]: It is an extension of the SimpleTOD model on the SIMMC dataset,

supporting multimodal inputs. It views system action prediction as a causal language modeling task and finetunes the pretrained GPT2 language model to generate both system actions and responses. (2) **Multi-Task Learning** [18]: It utilizes multitask learning techniques to train a GPT2-based model, demonstrating robust performance across all tasks on the SIMMC dataset. (3) **Encoder-Decoder** [12]: It is an end-to-end encoder-decoder model based on BART for generating outputs, achieving first place in the overall ranking in the SIMMC competition. (4) **Reasoner** [26]: It employs a multi-step reasoning method and performs exceptionally well in the SIMMC 2.0 competition. (5) **MiniGPT4** [47]: For this widely used multimodal LLM, we concatenate the dialogue history and scene snapshot as input for the model and view all three subtasks as response generation tasks to generate results. (6) **GPT-4o** [30]: It is a state-of-the-art multimodal LLM developed by OpenAI. To ensure a fair comparison, we follow the same setting as MiniGPT4 and adopt official configurations during inference.

### 5.2 Automatic Evaluation

The automatic evaluation results for the three subtasks on the SCREEN dataset are presented in Table 3, with the best metrics highlighted in bold. GPT4o received the highest score, as expected. Among the open-source models, MiniGPT4 outperformed other models across all subtasks, benefiting from its advanced language understanding and generation capabilities based on an LLM. In contrast, SimpleTOD+MM and MultiTask Learning, based on GPT2, showed weaker performance. The Encoder-Decoder and Reasoner models performed similarly, though the Reasoner had a slight edge due to its dual-system mechanism. Notably, all models struggled with situated recommendations, underscoring the challenge of capturing user preferences in specific scenarios. Even GPT-4o, while accurate in system action prediction, faced difficulties in recommending items and generating responses, two critical tasks in SCRS.

## 5.3 Human Evaluation

We engaged three well-educated annotators to manually evaluate the system-generated responses. To assess the relevance of these responses to the contextual scene, we introduced a novel metric, *Situated Relevance.* This metric evaluates whether the responses accurately reference items in the scene and consider the user's appearance and climate conditions. Additionally, we employed the criteria from [43, 45] to assess *Fluency* and *Informativeness.* Each indicator was scored on a scale from 0 to 2, where 0 indicates no relevance, informativeness, or fluency, and 2 signifies high relevance, rich information, and smooth fluency. To determine inter-annotator agreement, we calculated Fleiss's kappa [8] and aggregated the scores to derive the average human-evaluated results. The human evaluation results in Table 4 show that Fleiss's kappa scores are within the [0.4, 0.6] range, indicating moderate agreement among annotators. These findings closely align with the results from automatic evaluations, supporting the effectiveness of the three designed subtasks in assessing SCRS performance. Notably, GPT-4o and MiniGPT4 outperform other models in generating more situation-relevant, fluent, and informative responses. Although the Reasoner and the Encoder-Decoder models demonstrate comparable levels of situation relevance and fluency, the Reasoner's outputs are more informative due to its multi-step reasoning process that gathers necessary elements for generation.

We conducted a human evaluation to verify the reliability of the SCREEN dataset. We randomly selected 50 dialogues each from SCREEN and SIMMC 2.1, forming dialogue pairs, and asked five human evaluators to assess these pairs based on the following criteria: "Situation Relevance," which determines which dialogue is more relevant to the scene; "Personality," which evaluates which dialogue better reflects the user's personality; "User State," which assesses which dialogue considers the user's mood and appearance more; and "Rec. Appropriateness," which judges which dialogue's recommendation is more appropriate. The comparative results, presented in Figure 6, indicate that the SCREEN dataset achieves higher win percentages than the artificially generated SIMMC 2.1 dataset. This outcome demonstrates the reliability of our dataset.

## 5.4 Discussions

To demonstrate the quality of responses generated by those baseline models on the SCREEN dataset, we present an illustrative case in Figure 7. We observe that Reasoner and MiniGPT4 can successfully utilize contextual information from the scene (e.g., climate: summer) and conversational history (e.g., playing basketball) to make appropriate recommendations. While SimpleTOD+MM also attempts a recommendation, it fails to specify the clothing recommended. The MultiTask Learning and Encoder-Decoder models limit their outputs to mere descriptions, omitting recommendations. Significantly, MiniGPT4 demonstrates an enhanced ability to generate responses enriched with informative content, underscoring the advanced capabilities of LLMs. Nonetheless, we draw a conclusion that there is still a large room to improve these baseline models to fully address situated recommendations, remaining huge research potential in the future.

We also identify some limitations of this work as follows. Utilizing LLM agents to simulate predefined roles in developing SCRS

**Situated Conversational Recommendation System**

**Figure 7: Case study for different baseline models on the SCREEN dataset.**

still poses some challenges. Despite efforts to increase variability through controlled settings, LLMs occasionally generate responses with hallucinations [1]. In the future, post-processing measures such as verification and corrections by multiple moderators will be designed to enhance dataset quality. Moreover, rigorous ethical considerations are paramount, especially in preventing the generation of harmful content and ensuring no sensitive and private information should be involved. To some extent, this can be alleviated through manual sampling inspection.

## 6 Conclusion

This work proposes a novel problem setting named Situated Conversational Recommendation System (SCRS) that enhances traditional multimodal conversational recommendations by integrating situational factors. To facilitate advancements in this field, we construct a comprehensive, high-quality benchmark named SCREEN using an efficient role-playing approach based on multiple LLM agents. Furthermore, we define three essential subtasks for SCRS and evaluate several representative baseline models, moving to a new research direction that narrows the gap between traditional and real-world conversational recommendations.

## Acknowledgments

This work was supported by the National Natural Science Foundation of China (62076212), the Research Grants Council of Hong Kong (15207122, 15207920, 15207821), and PolyU internal grants (ZVQ0, ZVVX). The authors would like to thank the anonymous reviewers for their valuable feedback and constructive suggestions.

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
