# OpenReview forum: "SCREEN: A Benchmark for Situated Conversational Recommendation"
_acmmm.org/ACMMM/2024/Conference — MM2024 Poster_

### Official Review · Reviewer_DX2Q · 2024-05-22

**Rating:** 4
**Confidence:** 3

**Summary:**

The authors introduce a new Benchmark for Situated Conversational Recommendation Systems. Compared with the existing datasets, they leverage LLM agents via role-playing approach to simulate the environmental and situational context for both the user and system such as the climate, user personality, etc. They also propose three subtasks to evaluate the performance of the existing baselines.

**Strengths:**

1.	This is the first dataset specifically designed for situated conversational recommendation systems, addressing the gap in the current research landscape.
2.	The dataset incorporates a wider range of situational factors that influence user behavior in SCRS scenarios, providing a richer context for evaluation.
3.	The paper is well-organized and easy to follow.

**Limitations:**

1.	While the dataset includes various situational factors like climate and personality traits, it is unclear if these factors significantly influence users' purchase behaviors. The practical applications and the potential impact of the dataset for real-world scenario should be further discussed.
2.	There is a lack of detailed description regarding the measures taken to ensure the quality and accuracy of the generated dialogues. Additionally, more data statistical analysis is needed to help users understand the dataset comprehensively
3.	Most of the baseline models used are from the SIMMC dataset. It would be beneficial if the authors could design a simple, specific baseline tailored to the proposed dataset to better evaluate its unique aspects.
4.	It is not specified whether the authors will make the dataset publicly available.

**Suitability:**

3

---

### Official Review · Reviewer_PZP1 · 2024-05-25

**Rating:** 5
**Confidence:** 3

**Summary:**

This paper introduces Situated Conversational Recommendation Systems (SCRS) to address the impact of situational factors on recommendations. The paper employs large language models to perform multi-role simulations, thereby establishing a novel benchmark dataset based on the SIMMC dataset. The paper has a good structure and is easy to read. This paper provides a detailed description of the experimental procedure. Particularly, Figure 2 clearly illustrates the main steps involved in the dataset construction, facilitating the reproducibility of this study. Furthermore, the paper offers a comprehensive analysis of various tasks based on the proposed benchmark, including both model evaluation and human evaluation.

However, I have two concerns:
1. Since all the utterances in the dataset, including the Ground Truth, are generated by GPT-4, could there be an inherent bias favouring the ChatGPT series of large models when using this benchmark for evaluation?
2. Although the authors mention in Section 5.3 that the results of human evaluation are consistent with those of automatic evaluation, I think merely observing the ranking of models is insufficient. It is necessary to further validate the consistency between automatic evaluation metrics and human annotations on the new benchmark. For example, calculating the correlation between metric and human scores, or using Predictive Power, may help to assess the reliability of this benchmark.

**Strengths:**

1. This paper provides a detailed description of the experimental procedure.
2. Extensive analyses are conducted.
3. The paper has a good structure and is easy to read.

**Limitations:**

Please see the summary.

**Suitability:**

3

---

### Official Review · Reviewer_aekR · 2024-05-25

**Rating:** 3
**Confidence:** 2

**Summary:**

In this paper,  the author pioneering to address this gap and introduce a novel setting: Situated Conversational Recommendation Systems (SCRS). We observe an emergent need for high-quality datasets, and building one from scratch requires tremendous human effort. To this end, they construct a new benchmark, named SCREEN, via a role-playing method using large language models. This benchmark
comprises over 20k dialogues across 1.5k diverse situations, providing a rich foundation for exploring situational influences on conversational  recommendations. Based on the SCREEN, the author propose three worth-exploring subtasks and evaluate several representative baseline models. The evaluations confirm that the benchmark is high quality, establishing a robust experimental basis for future research in situated conversational recommendation.

**Strengths:**

The paper aims to construct a benchmark dataset for recommendation tasks that is more in line with real-life scenarios, as there is limited situational information in existing recommendation tasks. This dataset has certain data value for recommendation tasks.

**Limitations:**

1 Is the dataset that integrates scenarios compatible with real scenarios and is it universal for recommendation tasks.
2 The paper proposes multiple baseline methods for experimental validation of the dataset, proving that the dataset can measure the advantages and disadvantages of the methods, as well as data quality analysis. But has this dataset been publicly evaluated through evaluation competitions?
3 What are the advantages of analyzing the size of the dataset compared to existing recommendation task datasets, only by integrating scenarios? The comparative experiments conducted in the paper are not sufficient.

**Suitability:**

2

---

### Official Review · Reviewer_KJZx · 2024-06-06

**Rating:** 3
**Confidence:** 3

**Summary:**

This paper construct a new benchmark dataset for a novel task of situated conversational recommendation. It first proposes this new task formulation, then use LLMs as agents to simulate and build the dataset. Multiple baseline methods are implemented on this dataset.

**Strengths:**

1. The motivation of introducing situation into the task of conversational recommendation makes sense.
2. The idea of leveraging LLMs and agents to simulate and study the problem is reasonable and promising.
3. The overall formulation and system design of the role-playing (agent) system make sense.

**Limitations:**

1. The dataset is constructed using more advanced LLMs such as GPT-4V and gpt-3.5-turbo, while the strongest LLM-based baseline is MiniGPT4, which is much weaker than the LLMs used for dataset construction. I am wondering how such advanced LLMs perform in this dataset. I am worrying about that the user actions in the dataset are generated by LLMs, and a consistent and integrated LLM can predict the actions very well especially when the actions are generated by itself.
2. A follow-up question is there is no direct and systematic evaluation of the quality of the dataset. It is unclear to what degree that the generated dataset is aligned with real world scenarios. If such dataset deviates much from real use cases, it would be less meaningful or reliable to conduct study on this dataset.
3. There should be a formal definition of the situation. Since situation is the main focus of this work, it is necessary to give formal definition of this key concept, for example, what characterstics, properties, or functions should be conveyed by situation. Even though there are examples such as temproal, spatial, climate, etc, it lacks an overall comprehensive definiton.
4. Using agent to study the conversation recommender system is a good idea, however I respectfully disagree to use it as a dataset construction tool. Because LLMs are sufficiently powerful to solve such problems, and we can directly use them as solutions rather than restrict them to build datasets.

**Suitability:**

3

---

### Meta-Review · Area_Chair_xhuK · 2024-06-29

**Recommendation:** Accept (Poster)
**Confidence:** 4

**Metareview:**

The reviewers have different opinions on their evaluation of this article. After reading this paper, I think the paper meets MM standards, and lean towards acceptance.